# Geometric Self-Supervised Learning: A Novel AI Approach Towards Quantitative and Explainable Diabetic Retinopathy Detection

**DOI:** 10.3390/bioengineering12020157

**Published:** 2025-02-06

**Authors:** Lucas Pu, Oliver Beale, Xin Meng

**Affiliations:** 1Department of Radiology, University of Pittsburgh School of Medicine, Pittsburgh, PA 15260, USA; lucapu@pitt.edu; 2Department of Ophthalmology, University of Pittsburgh School of Medicine, Pittsburgh, PA 15213, USA; bealeo2@upmc.edu

**Keywords:** diabetic retinopathy, lesion-level detection, image segmentation, self-supervised learning, shape descriptor

## Abstract

Background: Diabetic retinopathy (DR) is the leading cause of blindness among working-age adults. Early detection is crucial to reducing DR-related vision loss risk but is fraught with challenges. Manual detection is labor-intensive and often misses tiny DR lesions, necessitating automated detection. Objective: We aimed to develop and validate an annotation-free deep learning strategy for the automatic detection of exudates and bleeding spots on color fundus photography (CFP) images and ultrawide field (UWF) retinal images. Materials and Methods: Three cohorts were created: two CFP cohorts (Kaggle-CFP and E-Ophtha) and one UWF cohort. Kaggle-CFP was used for algorithm development, while E-Ophtha, with manually annotated DR-related lesions, served as the independent test set. For additional independent testing, 50 DR-positive cases from both the Kaggle-CFP and UWF cohorts were manually outlined for bleeding and exudate spots. The remaining cases were used for algorithm training. A multiscale contrast-based shape descriptor transformed DR-verified retinal images into contrast fields. High-contrast regions were identified, and local image patches from abnormal and normal areas were extracted to train a U-Net model. Model performance was evaluated using sensitivity and false positive rates based on manual annotations in the independent test sets. Results: Our trained model on the independent CFP cohort achieved high sensitivities for detecting and segmenting DR lesions: microaneurysms (91.5%, 9.04 false positives per image), hemorrhages (92.6%, 2.26 false positives per image), hard exudates (92.3%, 7.72 false positives per image), and soft exudates (90.7%, 0.18 false positives per image). For UWF images, the model’s performance varied by lesion size. Bleeding detection sensitivity increased with lesion size, from 41.9% (6.48 false positives per image) for the smallest spots to 93.4% (5.80 false positives per image) for the largest. Exudate detection showed high sensitivity across all sizes, ranging from 86.9% (24.94 false positives per image) to 96.2% (6.40 false positives per image), though false positive rates were higher for smaller lesions. Conclusions: Our experiments demonstrate the feasibility of training a deep learning neural network for detecting and segmenting DR-related lesions without relying on their manual annotations.

## 1. Introduction

In 2021, approximately 537 million adults had diabetes worldwide [1]. Within 15 years of diagnoses, around 80% of all diabetic patients will develop an eye disease called diabetic retinopathy (DR) [2,3]. Notably, DR stands as the leading cause of blindness in working-age adults globally, affecting over 103 million people in 2020, and is projected to surpass 160 million by 2045. DR often does not have any symptoms in its early stages, but once symptoms appear, there is a considerable risk of permanent vision loss, even with treatment. Early detection through routine retinal screening can reduce the risk of DR-related vision loss by up to 90%, thus being pivotal in managing and preventing vision loss [4].

In practice, DR assessment is performed manually by ophthalmologists but is posed with several challenges. In DR’s early stages, tiny lesions like microaneurysms make manual detection very time-consuming to find them and they are often missed, causing DR to develop to its late stages. On the other hand, in DR’s moderate to severe stages, numerous lesions of various types are often present. This makes it nearly impossible to quantitatively assess the disease’s extent, which is crucial to facilitate precision medicine. Moreover, the scarcity of ophthalmologists worldwide for screening, coupled with the increasing prevalence of diabetes, exacerbates this challenge [5,6]. Given these issues, significant effort has been made to develop artificial intelligence (AI) systems for automated DR detection.

To date, most studies have focused on utilizing deep learning (DL) to automatically classify color fundus photography (CFP) images into different DR severities. Many of these developed algorithms have demonstrated promising classification performance, leading to the approval of several automated DR detection products, such as Google’s ARDA and Digital Diagnostics’ IDx-DR, by the Food and Drug Administration (FDA). However, despite their reported efficacy and FDA approval, their adoption in clinical practice remains limited [7]. This stark reality is primarily attributed to three reasons. First, existing automated systems only classify DR severity into two categories, with healthy and mild-DR-stage patients in one group. This leads to a failure to identify mild DR patients for a further diagnosis, preventing them from receiving attention and necessary treatments. Second, these systems’ sole classification outputs cannot provide valuable quantitative insight into DR to support precision medicine applications. Third, these systems face challenges with interpretability due to the “black-box” nature of deep learning, which can make clinicians hesitant to fully trust their predictions. As a result, existing automated detection systems have not yet effectively addressed the urgent challenges present in manual detection.

In contrast, a quantitative detection system that presents the precise extent, location, size, and type of DR lesions through segmentation has the potential to overcome the limitations in classification-based systems. This quantitative information can enable fine-grained DR progression monitoring that is essential for planning personalized treatments, observing the effects of treatments like anti-VEGF injections on DR lesions, and clinically validating new therapies. In addition, classifying DR severity based on quantitative measures like the count of lesions may enable more accurate prognostication [8].

Although there have been studies aiming to develop segmentation algorithms to explicitly identify individual DR-related lesions (Figure 1), such as microaneurysms [9], hemorrhages [10,11], and exudates [10,12], their performance remains unsatisfactory, with most focusing on only one or two types of DR lesions. These issues are rooted in the substantial reliance on manual annotations as ground truths, a time-consuming and challenging process that hinders the availability of extensive and diverse datasets with manually outlined DR lesions for training purposes. Numerous lesions of various types may be present, often taking hours to annotate a single image with severe disease. In addition, small lesions like microaneurysms that are only a few pixels large are often overlooked during manual outlining. This makes it hard to guarantee the accuracy of manual annotations, potentially leading to suboptimal performance of the trained AI models. Hence, although millions of fundus images have been accumulated in clinical practices, it becomes challenging to efficiently utilize them to develop an accurate and applicable segmentation model due to the lack of reliable ground truth annotations.

To address this issue, this study aims to develop and validate a novel self-supervised approach enabling any designated DL model to detect, segment, and differentiate the four most prevalent DR lesions (i.e., microaneurysms, hemorrhages, hard exudates, and soft exudates) without the need for manual annotations as ground truths. The underlying idea is to develop novel geometric computer vision and rule-based algorithms to accurately identify regions in the image with high confidence of being DR lesions. These regions serve as ground truths for training a convolutional neural network (CNN)-based model. Since no manual annotation is needed to generate ground truths, the CNN model can utilize the numerous amounts of images generated from the clinical setting to improve its performance with minimal manual interference. To validate the generalizability of the developed self-supervised learning approach, we also validated its performance on ultrawide field (UWF) retinal images. UWF is an emerging retinal imaging technology that provides a broader view of the retina than standard CFP images. A detailed description of the algorithm and its performance follows.

## 2. Methods and Materials

### 2.1. Study Cohorts

This study identified three cohorts from public sources to develop and validate the proposed algorithm. The first cohort, denoted as Kaggle-CFP, consists of 2100 CFPs from a public dataset on Kaggle [13]. Each CFP in this dataset was rated by an ophthalmologist for DR severity on a five-point scale, including normal, mild, moderate, severe, and proliferative. Among the 2100 CFPs, 1000 were rated as normal, and 1100 were rated from mild to severe but without proliferative DR. A total of 2050 cases in this cohort were used for algorithm development, with the remaining 50 CFPs verified with moderate and severe DR set aside as an independent test set to assess the performance of the algorithms. An ophthalmologist manually located and outlined microaneurysms, hemorrhages, hard exudates, and soft exudates on the retinal images in the independent test set. Among the DR-positive cases for algorithm development, 100 CFPs were further set aside for internal validation, with the remaining cases (*n* = 1950) used for training.

A public dataset called E-Ophtha [14] with manual annotations was used as the second independent test set to compare with available studies. E-Ophtha contains two subsets. The first one, E-Ophtha-Exudate, has 47 CFPs with manual annotations of exudates. The second one, E-Ophtha-MA, has 148 CFPs with manual annotations of microaneurysms. The images were acquired from more than 30 medical centers in France and demonstrate a variety of image quality, resolution, and illumination conditions.

We also collected a third cohort consisting of ultrawide field (UWF) retinal color fundus images to validate the generalizability of the developed algorithm. A total of 256 UWF images were acquired on diabetic patients using the Optomap P200Tx device (Optos, Dunfermline, UK) from another public dataset [15]. Of these UWF images, 182 were verified with DR. We split the cases in the UWF cohort into two groups for algorithm development and independent validation, respectively. The independent validation group contained 50 UWFs verified with moderate and severe DR to challenge the algorithms. The remaining cases were used for algorithm development. Among the DR-positive cases for algorithm development, we set aside 30 UWFs for internal validation, with the remaining cases used for training. An ophthalmologist manually located and outlined exudates and bleeding spots on the retinal images in the independent test groups.

The selection of public datasets allows a fair comparison with available methods based on the same datasets. For a consistent image process, we resized the CFP images to uniform dimensions of 1000 × 1000 pixels and the UWF images to uniform dimensions of 1500 × 1500 pixels as an image preprocessing step.

### 2.2. Algorithm Overview

The developed algorithm consists of the following steps (see Figure 2). First, a retinal image was transformed into a contrast field using a novel geometric computer vision algorithm called a “multiscale contrast shape descriptor (MCSD)” to locate and identify areas that might potentially be DR lesions. Second, based on this contrast field, the identified regions were scored to determine their confidence of being DR lesions based on their intensity and contrast with surrounding structures. Third, identified high-confidence regions were classified into hard/soft exudates and microaneurysms/hemorrhages according to their size, shape, and intensity as compared with their surrounding regions. Finally, image patches were extracted on them to train four U-Net segmentation models to automatically detect and outline the boundaries of each DR lesion type, respectively.

### 2.3. A Multiscale Contrast Shape Descriptor (MCSD)

We first identified potential DR lesions (i.e., microaneurysms, hemorrhages, hard/soft exudates) from CFP images. These lesions can be largely classified as blob-like structures with a somewhat isotropic appearance (see Figure 3). In addition, they display pixel intensities distinct from the background (i.e., retina). Consequently, by computing the absolute difference between the pixel intensities of the lesions and those of the local background pixels, a significant intensity contrast can be observed. Furthermore, considering their isotropic nature, if the mean pixel intensity along a line segment originating from a lesion pixel in any direction is compared with the original pixel intensity, a notable difference is expected (see dashed lines in Figure 3). However, retinal vessels, being a prominent landmark, also exhibit different pixel intensities compared to the background, introducing the possibility of them being erroneously identified as lesions. In contrast to blob-like structures, retinal vessels can be predominantly categorized as linear structures due to their elongated shape, displaying consistent pixel intensities along the direction of the vessels (see Figure 4). Exploiting this characteristic, if a line originates from a pixel within a vessel and extends in every direction, the minimum absolute difference consistently aligns with the vessel’s direction. However, this principle does not apply to blob-like structures due to their isotropic shape, thus differentiating between vessels and lesions. Based on these characteristics, a novel algorithm termed a “multiscale contrast shape descriptor (MCSD)” was developed to differentiate these structures. Mathematically, this shape descriptor can be represented as(1)$$C\left(i,j\right)=\left\{\begin{array}{c} linear\;shape:\;\max\{c_{\theta }=\left|\frac{1}{d}\sum_{k=1}^{d}I_{k}^{\theta }-I\left(i,j\right)\right|,\;\theta \in [0,2\pi ]\}\;\\ blob\;shape:\;\min\{c_{\theta }=\left|\frac{1}{d}\sum_{k=1}^{d}I_{k}^{\theta }-I\left(i,j\right)\right|,\;\theta \in [0,2\pi ]\}\; \end{array}\right.$$
where *C*(*i, j*) is the contrast level or intensity, after applying the shape descriptor; *d* is the length of a line segment starting from pixel point $I\left(i,j\right)$ (e.g., dashed lines and point *o* in Figure 3); *θ* is the angle of the line segment with respect to the horizontal direction; and $I_{k}^{\theta }$ is the intensity of each pixel on the line segment. *θ* determines the number of line segments and directions (see Figure 3). For example, *θ* = π would indicate two line segments. The length *d* of the line segments is tuned so a smaller *d* differentiates smaller structures and a larger *d* differentiates larger structures. Since bleeding spots (i.e., microaneurysms and hemorrhages) have lower intensities than the background, the difference between them and the mean of a line segment extended into the background would be positive. Since exudates have higher intensities than the background, this difference would be negative.

MCSD effectively detects small and large regions that might potentially be DR lesions by setting the length of the line segment at different levels. Notably, MCSD was able to effectively distinguish bleeding spots from neighboring vessels, even though the intensities of the bleeding spots were similar to those of the vessels, and the spots were often attached to the vessels. This is demonstrated in the examples shown in Figure 4.

### 2.4. Scoring of Identified Regions to Determine Their Confidence Levels

Because some identified regions might just be image noise, we then assessed each region’s confidence of being a DR lesion. To conduct this, we computed two rings surrounding each region: an inner ring to characterize the region’s intensity and an outer ring to characterize the intensity of its surrounding neighborhood (see Figure 5). The difference between these two rings is used to characterize the region’s confidence. A significant contrast indicates differing pixel intensities compared to the background, suggesting a higher confidence that it is a lesion. Specifically, confidence is quantified as the ratio of the absolute intensity difference between the two rings to the intensity of the inner ring:(2)$$Confidence\;Level=\frac{\left|outer-inner\right|}{inner}$$

A cutoff threshold of 0.15 of the confidence level was used to filter out the low-confidence regions, and a threshold of 0.35 was used to determine the high-confidence regions. Any potential regions with a confidence level between 0.15 and 0.35 were marked as indeterminate regions. The two rings were computed using the Fast Marching Method (FMM) to obtain a signed distance field, characterizing the minimum distance of each pixel to its closest region boundary. In this field, the points inside an identified region have positive distance values, the points outside the region have negative distance values, and the points on its boundary have values of zero. In our implementation, the width of the two rings around the centroid of the lesion was set to one-fourth of the maximum radius of the identified lesion region. The radius and centroid of the lesion can be directly obtained from the signed distance field, which is the maximum distance value within the spot and the pixel with this value, respectively. Notably, as shown in Figure 5b,c, bleeding spots (i.e., hemorrhages and microaneurysms) may be partially fused with retinal vessels due to their similar intensity (Figure 6B,C). To avoid incorrectly identifying these noisy regions as bleeding spots, we calculated the ratio between the area of the non-vessel part in the ring and the total area of the ring. If this ratio exceeds a specified threshold (e.g., 0.35), the associated bleeding spot is excluded from the bleeding spots (e.g., the example in Figure 6C).

### 2.5. Rule-Based Differentiation of DR Lesion Types

After identifying high-confidence regions (i.e., DR lesions), they were then differentiated into different lesion types. The first step was to largely differentiate them into two categories: exudates (i.e., soft and hard) and bleeding spots (i.e., microaneurysms and hemorrhages). Exudates and bleeding spots can be effectively distinguished from each other due to their opposite intensity contrast with the background. The second step was to differentiate bleeding spots into microaneurysms and hemorrhages. Microaneurysms typically appear as very small and round regions, while hemorrhages appear as large dimensions with irregular shapes. Based on this, a principal component analysis (PCA) was used to analyze the roundness or elongation of a DR lesion. Given the coordinates of the MCSD-identified lesion boundaries, PCA identifies two eigenvalues representing the magnitude of the variances. The ratio between them was used to determine the elongation of each isolated region since they represented the shape. A ratio close to 1 translates to a round region and, thus, a microaneurysm. In our implementation, a ratio threshold is set to be 1.25. Above this threshold translates to a hemorrhage. A size differentiation threshold is also set to be a microaneurysm if the lesion is less than 1/200th the size of the image.

The final step is to differentiate exudates into soft and hard types. Hard exudates generally have a sharp intensity and a progressive intensity decrease towards their boundaries. Soft exudates, on the other hand, have a more uneven intensity distribution, often resembling “cotton-wool” patterns. This “cotton-wool” pattern was used to distinguish them from hard exudates using the calculated inner ring described in Section 2.4. The mean pixel intensity of the lesion inside the calculated inner ring is compared to the mean pixel intensity of the lesion outside the inner ring. For soft exudates, due to their uneven distribution, a certain portion of the pixels in the outside lesion have pixel intensities greater than the mean intensity within the inner ring. This characteristic does not apply to hard exudates. Thus, a higher percentage of total outer pixels having greater intensities than the mean inner intensity means a higher uneven intensity distribution, translating to a soft exudate. In our implementation, this threshold is set to 15%.

### 2.6. Training a CNN-Based Segmentation for Detecting and Segmenting DR Lesions

The entire CFP image was searched to identify local low- or high-confidence image patches devoid of any indeterminate regions for learning backgrounds and those containing only high-confidence regions for learning individual lesions. The steps to execute this process include the following: (1) Starting with a high-confidence region or a background pixel, a small image patch (e.g., 10 × 10 pixels) centered on its centroid is initialized. (2) If there are no indeterminate regions within this patch, the size of the image patch is incrementally increased at a step, such as 5 × 5 pixels. If indeterminate regions are present or it reaches a maximum size (e.g., 120 × 120 pixels), this expanding procedure will stop. If the image patch size is larger than the predefined minimum (e.g., 60 × 60 pixels), this image patch will be used for the training procedure. To balance the training data, image patches are randomly extracted from retinal images without any DR lesions, namely DR-negative images, by following the same procedure. The high-confidence image patches and the image patches without any DR lesions are paired for training a DL model.

The U-Net model [16] was then trained based on the paired image patches to detect and segment the identified DR lesions, namely the high-confidence regions. The encoding and decoding paths of the U-Net were formed by four stages of convolution blocks, starting with 32 filters and doubling thereafter. Different U-Net models were trained for each type of abnormal finding: a hard exudate, soft exudate, microaneurysm, and hemorrhage. During training, the above image patch extraction procedure was performed on the fly, and various data augmentation operations were used to increase the diversity of the data and the reliability of the trained U-Net models. These operations included translation, cropping, padding, scaling, flipping, rotating, adding Gaussian noise, and shifting pixel intensities. The Dice coefficient was used as the loss function during training. The initial learning rate was set to 0.001 and reduced by a factor of 0.5 if the validation performance did not increase in three epochs. The training procedure was terminated when the validation performance did not improve compared to the previous ten epochs.

Considering the distinct characteristics of CFP and UWF images, we trained a U-Net model for the UWF image separately following the above training procedure.

### 2.7. Performance Evaluation

The performance of the developed U-Net models for CFP and UWF images was evaluated on their independent test sets, namely the Kaggle-CFP independent test set and the E-Ophtha dataset for the CFP images, and the UWF independent test set for UWF images. The sensitivity and false positive rate were used as performance metrics. An exudate or bleeding spot detected by the computer algorithm was considered a false positive if it did not overlap with manually outlined regions; otherwise, it was considered a true positive. These performance metrics were computed separately for hard exudates, soft exudates, microaneurysms, and hemorrhages depicted on CFP images. Due to the significantly broader field of view, inherent image distortion, poor image quality, and substantial artifacts in UWF images, accurately annotating different DR-related lesions is challenging. As a result, we did not categorize lesions as precisely as in traditional CFP images. However, we classified bleeding and exudate spots into more granular categories based on size.

## 3. Results

### 3.1. Validation on the Kaggle-CFP Independent Test Set

In the Kaggle-CFP independent test set, there were 797 microaneurysms, 1623 hemorrhages, 2319 hard exudates, and 86 soft exudates (Table 1). The developed U-Net models demonstrated a sensitivity ranging from 90.7 to 92.6%. The false positive detections per image ranged from 0.18 to 9.04. The examples in Figure 6, Figure 7 and Figure 8 show the performance of the U-Net models on CFP images.

### 3.2. Validation on the E-Ophtha Cohort

In the E-Ophtha-Exudate dataset (*n* = 47), there are 2278 isolated exudate spots manually detected and annotated by experts, with counts ranging from 1 to 253 across the CFP images. The proposed self-supervised U-Net model detected 2134 of these exudate spots, resulting in an overall sensitivity of 93.7%, irrespective of their size. In the E-Ophtha-MA dataset (*n* = 148), there are 1,306 isolated microaneurysms manually annotated by experts, with counts ranging from 1 to 48 across the CFP images. The U-Net model caught 1219 of these microaneurysms, achieving a sensitivity of 93.3%. Originally, there were 255 false positive exudate detections, namely 5.4 false positives per image, and 669 false positive microaneurysm detections, including the hemorrhages (Figure 8), amounting to an average of 4.5 false positives per image. The examples in Figure 8 and Figure 9 show the performance of the developed model on an image in the E-Ophtha dataset. It can be seen that in Figure 8, the developed self-supervised AI models detected multiple lesions that had even been overlooked in the original manual annotations. An ophthalmologist (>10 years’ experience) was asked to review both computer and original manual results to make necessary corrections. The false positive rate reduced from 5.4 to 2.85 per image for exudates and from 4.5 to 2.38 per image for microaneurysms. The developed models demonstrated much higher performance compared to available supervised models [11,17,18,19,20,21,22,23,24,25,26,27]. Table 2 lists the performance of the previous studies on the E-Ophtha dataset.

### 3.3. Validation on the UWF Independent Test Set

The CNN model trained on UWF images showed comparable sensitivity in detecting bleeding and exudate spots of larger sizes, but with significantly higher false positives compared to the model trained on CFP images (Table 3). This high rate of false positives was primarily due to the complex appearance of UWF images. Examples of high false positive detections of exudate and bleeding spots can be seen in Figure 10, where eyelashes were incorrectly detected as bleeding spots, and lens stains and laser burn spots were incorrectly detected as exudate spots.

Given a CFP or UWF image, it typically takes less than 10 s for the CNN models to identify the bleeding and exudate spots on a typical desktop with GPU and less than 30 s when only CPU is available.

## 4. Discussion

This self-supervised approach for DR lesion detection and segmentation utilizes the geometric features of DR lesions to generate a contrast field that serves as automatically generated ground truth masks for self-supervised training. Unlike most existing self-supervised learning methods, this novel approach makes it possible to utilize the vast number of clinical images available that lack human labels for training, and it eliminates the need for labor-intensive manual annotations. Differing from current systems that only provide image-level classification, the developed models in this study offer transparency by presenting precise information on lesion location, type, size, and extent. By explicitly identifying microaneurysms, the models can differentiate mild-stage DR patients from healthy patients for more precise diagnoses.

We note that the primary focus of this study was on devising a strategy to train a DL-based model without the need for manual annotations rather than optimizing the final segmentation model. In this pursuit, the classical U-Net architecture was trained without any modification or fine-tuning to avoid presenting higher performances than actual ones. Nonetheless, the U-Net model developed in this study exhibited superior performance on the E-Ophtha dataset in comparison to all existing studies that developed novel deep learning architectures (Table 2). If the proposed strategy was applied to train a more sophisticated algorithm, there would be significant improvements in performance.

The significance of the self-supervised approach extends beyond the efficient utilization of the millions of retinal images available in clinical practice to enhance performance but also addresses the potential errors and biases associated with manual annotations. To illustrate, a single image often contains numerous isolated small abnormal findings scattered throughout, rendering manual annotations a challenging and imprecise task. As demonstrated in Figure 9, in our opinion, the manual annotations frequently overlook a large portion of abnormal findings, including exudates and microaneurysms. Relying on these inaccurate ground truths hinders the development of a robust and accurate model. Instead, the developed self-supervised approach leverages inherent image characteristics to generate high-confidence image patches for training, allowing for unified ground truths that are not subjective to human error.

There could be several reasons for the false positives in the Kaggle-CFP independent test set. First, the images in the independent datasets vary significantly in image quality and have many other abnormal findings. Second, due to the large number and variety of abnormal findings, it is challenging for a human expert to fully annotate all of these findings, especially for challenging cases. Although a significant amount of time was dedicated to manually annotating 50 CFPs in the independent test sets, it is difficult to ensure their complete accuracy, especially for the small abnormal findings.

To evaluate the generalizability of the developed algorithm, we applied it to detect bleeding and exudate spots on UWF images. However, we observed high false positive rates for these detections. Several factors may contribute to this issue. First, the dataset of UWF images was relatively small. Second, while UWF images provide a panoramic view of the retina, they have notable weaknesses compared to traditional CFP images, including lower resolution, distortion in peripheral regions, and susceptibility to artifacts such as eyelid or eyelash shadows. These limitations hinder the detection of high-confidence regions, affecting the algorithm’s overall performance. Despite these challenges, the algorithm still showed promising results, suggesting the potential generalizability of the algorithm. Its performance could be significantly improved with a larger dataset and techniques to mitigate artifacts like shadows. We believe that the use of public datasets is a strength of this study, as it allows for fair comparisons with similar studies using the same datasets (Table 2).

Despite the proposed approach’s effectiveness and potential, there are some limitations in this study. First, there are no large datasets available for validation due to the challenges of manual annotation. As a result, although the independent test sets include images of various qualities collected from different medical centers, they only include a few hundred images. Second, we focus primarily on detecting and segmenting the most common DR-related lesions. However, other important lesions, such as venous beading, intraretinal microvascular abnormalities (IRMAs), and neovascularization, also play a significant role in DR progression. Identifying these lesions requires additional effort beyond the scope of this study and may be explored in our future work. Third, we do not validate the classification performance of our algorithm at the image level because the primary focus of this study is on validating the feasibility of detecting individual lesions without relying on manual annotations. According to the available guidelines [28,29], a screening program for DR should ideally have a sensitivity of at least 80% to accurately identify individuals with the disease and a specificity of at least 95% to reliably exclude those who do not have diabetic retinopathy. These performance criteria are based on image levels. In the future, we will explore and validate the potential clinical utility of our algorithms in screening settings.

Given these limitations, future work will involve creating a large test set with manual annotations to validate the models, potentially in the screening setting. Moreover, the superior performance of the models was primarily attributed to training on over 2000 images, facilitated by the elimination of manual annotations, whereas existing studies were only capable of training on a few hundred labeled images. As such, the next main step would be to utilize more images for training, preferably hundreds of thousands of images, which would further boost the models’ performance to another level. Since this study’s focus was on developing a novel strategy to facilitate self-supervised learning, the U-Net model was trained as a baseline performance measure. As a result, if more sophisticated architectures are implemented to develop an applicable quantitative system, the performance will undoubtedly increase. After performing these procedures to optimize the algorithm’s performance, we will explore its clinical utilities in real-world settings and its integration with an available clinical workflow.

## 5. Conclusions

This study developed and validated the first self-supervised approach for DR lesion detection and segmentation that can train a deep learning model without the need for any manual annotations as ground truths. It can detect and differentiate the four most prevalent DR lesions (i.e., microaneurysms, hemorrhages, soft exudates, and hard exudates) on color fundus images. The proposed approach leverages a novel computer vision algorithm called the “multiscale contrast shape descriptor (MCSD)” to extract local image patches that exhibit high-confidence DR lesion findings. These patches are then input into a U-Net model for precise image segmentation. The models were extensively validated on two public datasets and outperformed available studies in terms of sensitivity and false positive rate metrics. By eliminating the time-consuming and costly process of manual annotations, it is made possible to incorporate the millions of retinal images available from screening that are currently incapable of being used for training. This can enable the development of a robust and applicable quantitative DR detection system that is currently impossible to develop.

## Figures and Tables

**Figure 1 bioengineering-12-00157-f001:**
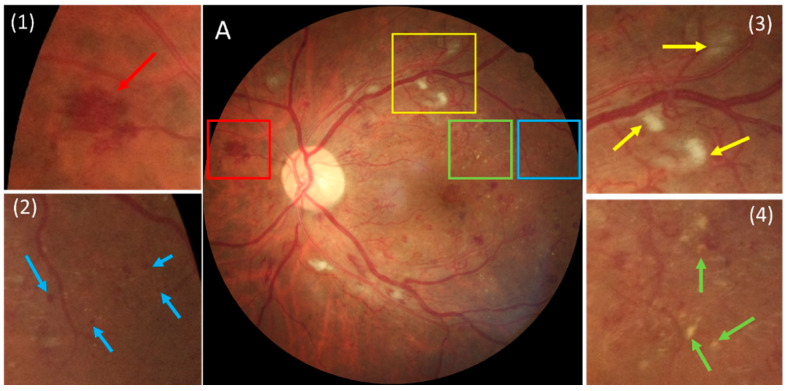
A color fundus photography (CFP) example illustrating the appearance of a hemorrhage, microaneurysm, soft exudate, and hard exudate. (**A**) shows the original image with boxed regions, while (**1**)–(**4**) present enlarged views. In (**1**), a red arrow indicates a hemorrhage, corresponding to the red box in (**A**). In (**2**), a blue arrow marks a microaneurysm from the blue box in (**A**). In (**3**), a yellow arrow points to a soft exudate from the yellow box in (**A**). In (**4**), a green arrow highlights a hard exudate from the green box in (**A**).

**Figure 2 bioengineering-12-00157-f002:**
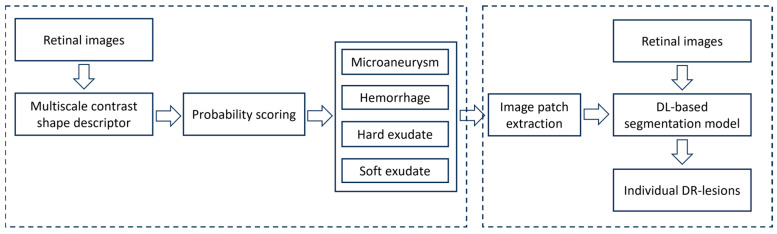
Overview of proposed geometric self-supervised approach. Stage 1 is based on geometric computer vision and rule-based algorithms, and Stage 2 is based on deep learning algorithms trained on identified image patches from Stage 1.

**Figure 3 bioengineering-12-00157-f003:**
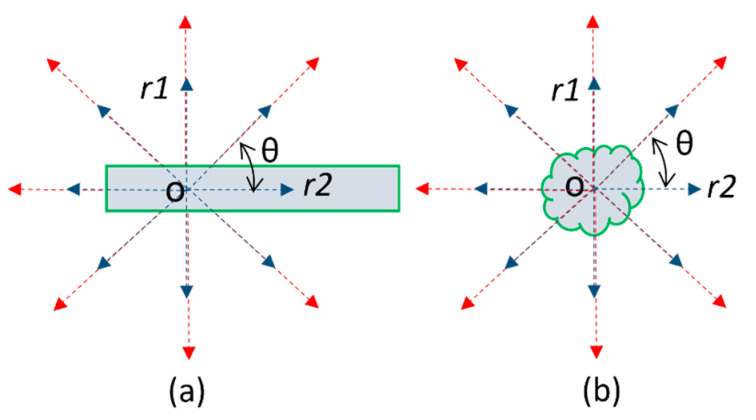
Illustration of multiscale contrast shape descriptor: (**a**) Linear structure and (**b**) blob-like structure. The different colors of the arrows represent varying distances of the line segments.

**Figure 4 bioengineering-12-00157-f004:**
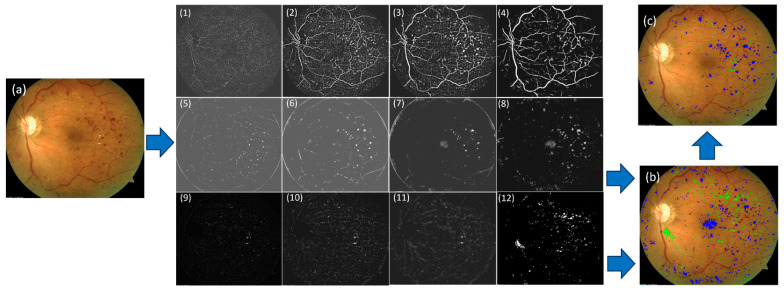
The results showcase the application of the MCSD to a CFP image in (**a**). (**1**)–(**3**): Results of the vessel descriptor applied to (**a**) with line segment lengths at scales 3, 5, and 10 pixels. (**5**)–(**7**): Results of the blob descriptor applied to (**a**) at scales 3, 5, and 10 pixels for detecting regions with a lower intensity than their background (i.e., hemorrhages and microaneurysms). (**9**)–(**11**): Results of the blob descriptor applied to (**a**) at scales 3, 5, and 10 pixels for detecting regions with a higher intensity than their background (i.e., exudate). (**4**), (**8**), and (**12**) show the merged results of the corresponding row after filtering low-contrast regions. (**b**) shows the merged results of (**8**) and (**12**) in an overlay. (**c**) shows the merged results in an overlay when thresholding to filter noisy regions based on their contrast values to the results in (**b**). Blue: bleeding spot (i.e., hemorrhage and microaneurysm) candidates. Green: exudate candidates.

**Figure 5 bioengineering-12-00157-f005:**
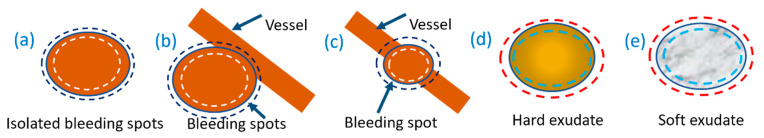
An illustration of the underlying idea of the boundary contrast for characterizing bleeding and exudate spots. (**a**) shows an isolated bleeding spot, (**b**) shows a bleeding spot slightly attached to a vessel. (**c**) shows a bleeding spot fused with a vessel. (**d**) shows a hard exudate that has a progressive density change from the center to the boundary. (**e**) shows a soft exudate that has an uneven density distribution. For a blob-like spot, both inner and outer rings as dashed lines are computed using EDT. The contrast between inner and outer rings is used to differentiate exudates and bleeding spots and quantify their confidence level.

**Figure 6 bioengineering-12-00157-f006:**
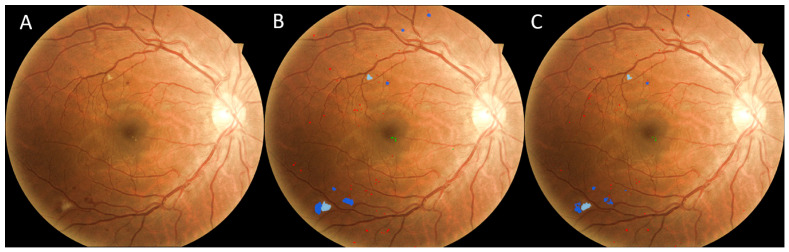
An example illustrating the performance of the U-Net models. (**A**) Original image, (**B**) manual result, (**C**) computer result. (Dark blue: hemorrhage, red: microaneurysm, green: hard exudate, light blue: soft exudate).

**Figure 7 bioengineering-12-00157-f007:**
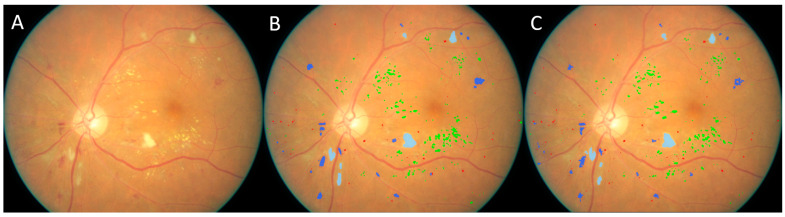
Another example demonstrating the models’ performance, successfully identifying tiny lesions of various types. (**A**) Original image, (**B**) manual result, (**C**) computer result. (Dark blue: hemorrhage, red: microaneurysm, green: hard exudate, light blue: soft exudate).

**Figure 8 bioengineering-12-00157-f008:**
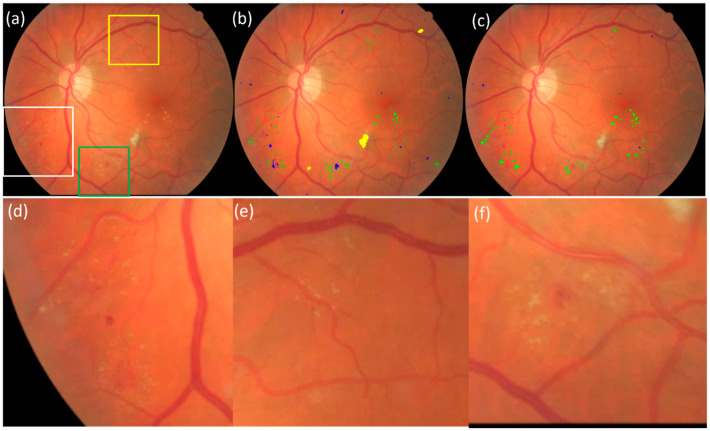
An example showing the performance of the U-Net models on the E-Ophtha dataset. (**a**) The original CFP image, (**b**) the computerized result by the U-Net models developed in this study, (**c**) the manual results, and (**d**–**f**) the local enlargement of the regions indicated by the boxes in white, yellow, and green, respectively. The manual annotation in (**b**) missed many lesions shown in (**d**–**f**) but they were detected by the developed models. The spots in green in (**b**,**c**) indicate the exudate. The spots in blue in (**b**,**c**) indicate the bleeding spots (including microaneurysm).

**Figure 9 bioengineering-12-00157-f009:**

Another example showing the models’ results. (**A**) Original image, (**B**) manual result, (**C**) computer result. Blue: bleeding spot, green: hard exudate, yellow: soft exudate.

**Figure 10 bioengineering-12-00157-f010:**
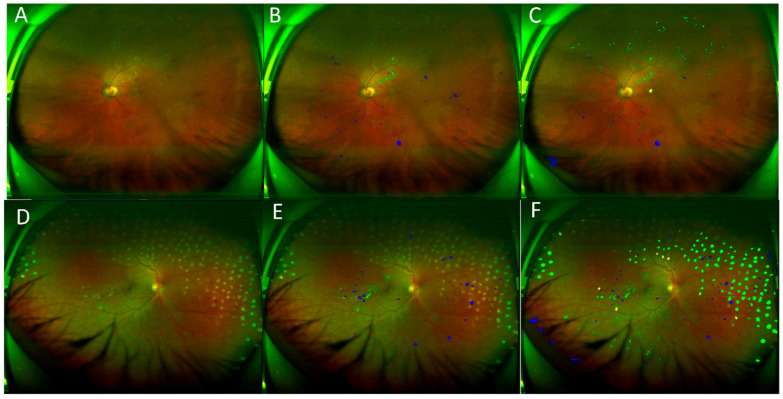
Examples with many false positive bleeding and exudate detections on UWF images. (**A**,**D**) Original image, (**B**,**E**) manual result, (**C**,**F**) computer result. Blue: bleeding spot, green: exudate.

**Table 1 bioengineering-12-00157-t001:** Performance of the U-Net models on the CFP independent test set (*n* = 50) for detecting and segmenting different types of DR lesions.

DR Lesions	Count in Manual Results	Sensitivity	False Positives (Per Image)
Microaneurysm	797	91.5% (729/797)	9.04 (452/50)
Hemorrhage	1623	92.6% (1503/1623)	2.26 (113/50)
Hard exudates	2319	92.3% (2140/2319)	7.72 (386/50)
Soft exudates	86	90.7% (78/86)	0.18 (9/50)

**Table 2 bioengineering-12-00157-t002:** The performance comparison of the AI models developed in this study based on the proposed geometric self-supervised learning and those in previous studies on the E-Ophtha dataset. Correction * means after a second review of the manual/computer segmentation results by our ophthalmologist.

Methods	Exudates		Microaneurysms	
	Sensitivity	False Positives	Sensitivity	False Positives
(Liu et al., 2017) [21]	0.760	–	–	–
(Zhang et al., 2014) [22]	0.740	–	–	–
(Fraz et al., 2017) [23]	0.812	–	–	–
(Das et al., 2017) [24]	0.858	–	–	–
(Piotr et al., 2018) [25]	0.846	–	–	–
(Dashtbozorg et al., 2018) [17]	–	–	0.638	8.0
(Piotr et al., 2018) [26]	–	–	0.621	8.0
(Melo et al., 2020) [18]	–	–	0.598	8.0
(Eftekhari et al., 2019) [20]	–	–	0.771	8.0
(Sun et al., 2021) [19]	–	–	0.835	8.0
(Wu et al., 2017) [27]	–	–	0.573	8.0
(Orlando et al., 2018) [11]	–	–	0.632	8.0
Our method without correction *	0.937	5.4	0.933	4.5
Our method with correction *	0.940	2.85	0.946	2.38

**Table 3 bioengineering-12-00157-t003:** The performance of the CNN models on the UWF independent test set (*n* = 50).

Abnormal Findings	Spot Area (Pixels)	Count in Manual Results	Sensitivity	False Positives (Per Image)
Bleeding	(0,10)	124	41.9% (52/124)	6.48
	[10,50)	457	68.5% (313/457)	10.74
	[50,100)	179	84.9% (152/179)	2.12
	[100,∞)	137	93.4% (128/137)	5.80
Exudates	(0,10)	252	86.9% (219/252)	24.94
	[10,50)	195	95.9% (187/195)	37.10
	[50,100)	28	92.9% (26/28)	33.96
	[100,∞)	26	96.2% (25/26)	6.40

## Data Availability

The datasets used in this study are publicly available and sourced from Kaggle-CFP, E-Ophtha, and the DeepDRiD dataset.

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
