# Peer review of "Geometric Self-Supervised Learning: A Novel AI Approach Towards Quantitative and Explainable Diabetic Retinopathy Detection"

_bioengineering, 2025, doi:10.3390/bioengineering12020157_

Round 1

Reviewer 1 Report

Comments and Suggestions for Authors

Geometric Self-Supervised Learning is a new method for diabetic retinopathy screening and offers greater sensitivity than other AI models. It also allows us to get out of the black box, which will facilitate easier acceptance by ophthalmologists.

1. Give the abstract the following structure: Introduction and objectives, material and methods, results and conclusion.

2. Introduction: Line 35. Change 80% to 60%: Within 15 years of diagnosis, around 80% of all diabetic patients will develop an eye disease called diabetic retinopathy.

3. Results. If I had 1000 images of the control group, it should have also given the following results: sensitivity, specificity and acuity.

4. Under discussion, add something similar: A screening program to be validated by ophthalmologists must have at least ≥ 80% sensitivity to diagnose those who have diabetic retinopathy and ≥ 95% specificity to exclude those who do not have diabetic retinopathy.

Reviewer 2 Report

Comments and Suggestions for Authors

The manuscript "Geometric Self-Supervised Learning: A Novel AI Approach Towards Quantitative and Explainable Diabetic Retinopathy Detection" introduces an innovative framework for detecting and segmenting diabetic retinopathy (DR) lesions using self-supervised learning. This method eliminates the need for manual annotations by leveraging geometric computer vision and rule-based algorithms to generate high-confidence training data. It has shown promising results in detecting four common DR-lesion types with high sensitivity and fewer false positives compared to existing models. However, there are significant areas for improvement in the study.

Major Comments:

  1. While the manuscript demonstrates the effectiveness of the proposed self-supervised learning approach, it lacks sufficient comparison with alternative methodologies, particularly supervised learning models beyond the U-Net architecture. This limits the reader's understanding of the broader applicability and impact of the proposed method.
  2. The manuscript claims improved detection of DR-lesions but does not provide detailed statistical analyses of performance variations across lesion types (e.g., microaneurysms vs. hemorrhages). Including more granular data would strengthen the conclusions.
  3. The selection criteria for the datasets and the preprocessing steps for images are inadequately explained. A clearer description of how datasets were curated, including challenges with varying image quality, would add transparency.
  4. The manuscript briefly mentions the applicability of the self-supervised approach to other medical imaging tasks but does not provide specific examples or validation in different clinical contexts. This undermines the generalizability claim.
  5. There is no discussion on the potential clinical integration of the proposed system, such as workflow implications, user interface design, or validation in real-world settings.
  6. The study's reliance on two public datasets (Kaggle-CFP and E-Ophtha) raises concerns about dataset diversity. Validation on additional datasets with different characteristics (e.g., ethnic variations, imaging equipment) is recommended.
  7. The limitations section is minimal and does not critically evaluate challenges such as false positives in regions with high vessel density or potential biases introduced by the geometric feature extraction method.
  8. Figures demonstrating results could benefit from higher resolution and clear annotations to make the findings more accessible to the reader.

Minor Comments:

  1. The introduction contains grammatical errors, such as "filled with challenges" (line 12), which could be rephrased to "fraught with challenges" for better readability.
  2. Abbreviations like "CFP" (Color Fundus Photography) are used without a clear initial definition, potentially confusing readers unfamiliar with the term.
  3. Figures lack consistent formatting, and some captions (e.g., Figure 6) are unclear about what is being compared.
  4. The manuscript switches between "U-Net" and "U-net" inconsistently. Standardizing the terminology would improve readability.
  5. References to prior studies are not always comprehensive, such as missing citations for alternative segmentation algorithms that could provide context for the novelty of the approach.
  6. Table 1 does not adequately explain performance metrics like sensitivity and false positives per image for non-expert readers.
  7. Equations presented in the methodology (e.g., for MCSD) are not sufficiently explained, especially regarding parameter choices.
  8. The keywords section contains overly broad terms like "self-supervised" without specifying their relevance to DR detection.
  9. Sentence structures in the abstract are overly complex, reducing clarity. Simplifying key points would enhance accessibility.
  10. The acknowledgment section incorrectly places funding information, which should be integrated with the methodology or discussion.

Reviewer 3 Report

Comments and Suggestions for Authors

This paper presents a new method to detect and segment diabetic retinopathy lesions like microaneurysms and hemorrhages without needing manual labeling. The new methods refers to computer vision and U-Net models to improve diagnosis and treatment precision.

The study is interesting, but some issues require further attention.

1. I am not aware if the utilized validation dataset allows generalizability. Even though the independent test sets included images from various medical centers, the total number of images was only a few hundred. I feel that this might limit the wider application of the study findings.

2. Why other types of lesions that can be found in proliferative-stage (e.g., neovascularization) were not included in the detection ?

3. How are you sure that the utilized model (with the standard U-Net architecture) exhibited the best performance? Did you test other models or new fine tuning of this model?

4. The proposed approach relies on geometric and rule-based algorithms. How does it perform on diverse data, handle noise, and address the possibility of false positive findings?

Round 2

Reviewer 3 Report

Comments and Suggestions for Authors

Thank you for your collaboration.

I have no further comments to make.